# Is Routine Prophylaxis Against *Pneumocystis jirovecii* Needed in Liver Transplantation? A Retrospective Single-Centre Experience and Current Prophylaxis Strategies in Spain

**DOI:** 10.3390/jcm9113573

**Published:** 2020-11-06

**Authors:** José Ignacio Fortea, Antonio Cuadrado, Ángela Puente, Paloma Álvarez Fernández, Patricia Huelin, Carmen Álvarez Tato, Inés García Carrera, Marina Cobreros, María Luisa Cagigal Cobo, Jorge Calvo Montes, Carlos Ruiz de Alegría Puig, Juan Carlos Rodríguez SanJuán, Federico José Castillo Suescun, Roberto Fernández Santiago, Juan Andrés Echeverri Cifuentes, Fernando Casafont, Javier Crespo, Emilio Fábrega

**Affiliations:** 1Gastroenterology and Hepatology Department, University Hospital Marqués de Valdecilla, 39008 Santander, Spain; antonio.cuadrado@scsalud.es (A.C.); angelam.puente@scsalud.es (Á.P.); palomaalvrz@gmail.com (P.Á.F.); patricia.huelin@scsalud.es (P.H.); carmenalvtato@hotmail.com (C.Á.T.); ines.garciac@scsalud.es (I.G.C.); marina.cobrerosdel@scsalud.es (M.C.); fcasafont@gmail.com (F.C.); javiercrespo1991@gmail.com (J.C.); emilio.fabrega@scsalud.es (E.F.); 2Group of Clinical and Translational Research in Digestive Diseases, Health Research Institute Marqués de Valdecilla (IDIVAL), 39011 Santander, Spain; 3Biomedical Research Networking Center in Hepatic and Digestive Diseases (CIBERehd), 28029 Madrid, Spain; 4Department of Pathological Anatomy, University Hospital Marqués de Valdecilla. 39008 Santander, Spain; mluisa.cagigal@scsalud.es; 5Department of Microbiology, University Hospital Marqués de Valdecilla, 39008 Santander, Spain; jorge.calvo@scsalud.es (J.C.M.); carlos.ruizdealegria@scsalud.es (C.R.d.A.P.); 6Department of General Surgery, University Hospital Marqués de Valdecilla, 39008 Santander, Spain; juancarlos.rodriguezs@scsalud.es (J.C.R.S.); federicojose.castillo@scsalud.es (F.J.C.S.); roberto.fernandez@scsalud.es (R.F.S.); juanandres.echeverri@scsalud.es (J.A.E.C.)

**Keywords:** Prophylaxis, *Pneumocystis jirovecii*, liver transplantation

## Abstract

In liver transplant (LT) recipients, *Pneumocystis jirovecii* pneumonia (PJP) is most frequently reported before 1992 when immunosuppressive regimens were more intense. It is uncertain whether universal PJP prophylaxis is still applicable in the contemporary LT setting. We aimed to examine the incidence of PJP in LT recipients followed at our institution where routine prophylaxis has never been practiced and to define the prophylaxis strategies currently employed among LT units in Spain. All LT performed from 1990 to October 2019 were retrospectively reviewed and Spanish LT units were queried via email to specify their current prophylaxis strategy. During the study period, 662 LT procedures were carried out on 610 patients. Five cases of PJP were identified, with only one occurring within the first 6 months. The cumulative incidence and incidence rate were 0.82% and 0.99 cases per 1000 person transplant years. All LT units responded, the majority of which provide prophylaxis (80%). Duration of prophylaxis, however, varied significantly. The low incidence of PJP in our unprophylaxed cohort, with most cases occurring beyond the usual recommended period of prophylaxis, questions a one-size-fits-all approach to PJP prophylaxis. A significant heterogeneity in prophylaxis strategies exists among Spanish LT centres.

## 1. Introduction

*Pneumocystis jirovecii*, formerly *Pneumocystis carinii*, is a ubiquitous, opportunistic fungus that causes *Pneumocystis jirovecii* pneumonia (PJP) in immunocompromised individuals, including solid organ transplant (SOT) recipients. This infection leads to substantial morbidity and mortality and, prior to the broad implementation of prophylaxis, the risk of developing PJP among SOT recipients was approximately 5–15% [1]. This figure exceeds the recommended incidence threshold of 3–5% for using prophylaxis [2] and, accordingly, current guidelines recommend anti-PJP prophylaxis for at least 6–12 months for all SOT recipients due to the higher degree of immunosuppression during these first months [1,3,4,5]. For lung and small bowel transplant recipients requiring higher intensity of immunosuppression or in case of prior PJP infection or chronic cytomegalovirus infection, guidelines recommend considering prolonged prophylaxis [1]. Trimethoprim-sulfamethoxazole (TMP-SMX) is the prophylactic drug of choice with two meta-analyses reporting a reduction in the risk of PJP occurrence of 85–91% in non-human immunodeficiency virus immunocompromised patients when compared to no prophylaxis [2,6].

The evidence supporting the use of anti-PJP prophylaxis in liver transplant (LT) recipients, however, is less clear. PJP incidence varies with the type of organ transplanted, the geographic region, the immunosuppressive regimen utilized, and the period studied [1,7]. The high incidences of PJP in the absence of prophylaxis reported in LT cohorts from the 1980s [8,9] contrast with those from recent series in which PJP incidence is below 3% [10,11,12,13,14] and even similar to incidences from LT recipients using prophylaxis (Table 1) [15,16,17,18,19,20,21,22,23,24,25]. Moreover, only one study concerning LT patients was included in the two meta-analyses reporting the efficacy of TMP-SMX prophylaxis, and this randomized clinical trial did not include a control group without prophylaxis as it assessed the efficacy and safety of weekly sulfadoxine/pyrimethamine compared with daily TMP-SMX [18]. These data question the risk–benefit ratio of a systematic PJP prophylaxis in LT recipients and may lead to variability in prophylactic strategies among centres. Few data are available in this latter regard and, to our knowledge, are restricted to the paediatric SOT setting [26,27].

In this report, we aim to examine the incidence and characteristics of PJP in LT recipients followed at our transplant centre where routine prophylaxis has not been practiced since the beginning of our LT program in 1990, and to define the prophylaxis strategies currently employed for PJP prevention among LT units in Spain.

## 2. Experimental Section

### 2.1. Patients

The Marques de Valdecilla University Hospital (Santander, Cantabria, Spain) is an urban, academic tertiary care centre with great expertise in organ transplantation. We conducted a retrospective review regarding PJP infection of all LT performed at our institution since the beginning of our adult LT program in November 1990 to October 2019. In the initial years, our centre performed all LT, not only from Cantabria, but also from several other Spanish autonomous communities such as Galicia, Basque Country, Canary Islands, Asturias, La Rioja and Castile and Leon. Over the following 12 years these regions progressively developed their own LT programs, and since 2009 our program has been responsible for all LT performed in Cantabria and La Rioja. The organ donation activity in these two Spanish autonomous communities is the highest of our country (above 80 donors per million of population) and, as of January 1, 2019, their combined population was 895,212 inhabitants. All patients received an ABO-compatible primary orthotopic LT from deceased donors using the piggyback operation [28] and no prophylaxis against PJP was undertaken, except for some patients with combined liver-kidney transplant.

In order to evaluate the local prevalence of PJP infection in other solid organ transplant (SOT) recipients at our institution, a retrospective review regarding this infection was also conducted in recipients of kidney (KT), heart (HT), and lung transplantation (LuT). Universal anti-PJP prophylaxis with TMP-SMX for 6 months has been indicated in all KT recipients since 1996, while this prophylaxis was prolonged for life in all LuT recipients since the beginning of the program. In contrast, the HT program has never applied prophylaxis against PJP.

### 2.2. Cases

PJP cases were defined by the following criteria: (1) new onset of respiratory symptoms; (2) radiological findings consistent with PJP infection; (3) microbiological demonstration of PJP infection (i.e., real-time quantitative PCR, and/or Grocott methenamine silver stain performed in samples from bronchial alveolar lavage (BAL), sputum (spontaneous or induced), and transbronchial/open lung biopsy) (Figure 1). Of note, PJP testing has always been systematically performed on BAL from LT recipients. Grocott methenamine silver stain was the standard method used at the beginning of the program, but hereinafter, PJP testing also routinely included PCR. In contrast, (1,3)-β-d-glucan detection is rarely used in the LT setting at our centre. Cases without microbiological confirmation were also included if the clinical (fever ± respiratory symptoms) and radiological picture (fine, bilateral, perihilar, diffuse infiltrates that progress to an interstitial alveolar butterfly pattern) supported the diagnosis of PJP. Information on demographics, indication for LT, time period between LT and PJP, diagnostic method, clinical presentation, treatment and outcome of PJP, co-existing infections, immunosuppressive regimens used at PJP diagnosis, and previous acute or chronic rejection were retrieved for all LT patients.

The identification of PJP cases was performed using three approaches: (1) individual review of the medical records of each LT recipient; (2) list of all laboratory-confirmed PJP cases from the Department of Microbiology; (3) hospital discharge records. The latter consisted of a list of all patients admitted to our hospital with diagnosis upon discharge of PJP registered as code 136.3 of the International Classification of Diseases, Ninth Revision, Clinical Modification (ICD-9-CM), and as code B59 or J17.3 of the ICD-10 (this replaced ICD-9-CM from January 2016). These microbiological and discharge records were cross-referenced by medical record number against a secure intramural database of all LT recipients transplanted at our centre. The search of PJP cases in the other SOT recipients did not include the individual review of their medical records and was limited to data obtained from the microbiological and discharge records, and also from each SOT database.

### 2.3. Immunosuppressive Drug Regimens and Cytomegalovirus Prophylaxis Protocol in Liver Transplantation

From 1990 to 1999, postoperative immunosuppression was based on triple therapy with cyclosporine A, azathioprine, and steroids. In subsequent years, tacrolimus replaced cyclosporine as the first-line therapy due to its better long-term graft and patient survival [29]. Similarly, mycophenolate mofetil (MMF) replaced azathioprine as the antimetabolite agent of choice and was generally used for treatment of T-cell-mediated rejection (TCMR) and/or for patients who had renal dysfunction limiting the dose of tacrolimus. The remaining patients received dual therapy with tacrolimus and steroids. The latter were tapered slowly during the first year at the beginning of the program and hereinafter were discontinued 3–6 months post-LT, except for those patients at higher immunological risk (e.g., immune-mediated diseases such autoimmune hepatitis). Since 2008, inhibitors of the mammalian target of rapamycin (mTORi) were generally used in case of intolerance to MMF and/or development of de novo malignancy after LT. In the last decade, induction therapy with the interleukin-2 receptor blockers (basiliximab) was given as a calcineurin-sparing agent to patients with prior or postoperative significant renal impairment (i.e., creatinine clearance <60 mL/min). In contrast, induction therapy with antithymocyte globulin has never been used. This is also the case for desensitization, since all patients received an ABO-compatible primary orthotopic LT. Long-term immunosuppression was adjusted to the recipient characteristics, etiology of primary liver disease, and magnitude of alloimmune activation, with the aim of minimizing immunosuppression as much as possible. In the event of moderate and severe TCMR, management consisted of pulses of steroids (typically 1 g of methylprednisolone daily for 3 days) and an increase in calcineurin inhibitor therapy with or without addition of other agents (antimetabolites or mTORi). Mild TMCR was generally treated by increasing calcineurin inhibitor therapy.

As far as the cytomegalovirus (CMV) prophylaxis protocol is concerned, our CMV-seronegative recipients who receive an organ from a CMV-seropositive donor (D+/R−) receive antiviral prophylaxis with valganciclovir 900 mg po once daily for 3–6 months. This drug is started within the seventh and tenth day after LT. Pre-emptive therapy with valganciclovir (900 mg po b.i.d. in recipients with normal renal function) is used instead in CMV R+ patients. In these LT recipients CMV viral load (quantitative nucleic acid testing) is measured weekly until discharge and then once every two weeks for the first three months. The viral threshold we use to initiate pre-emptive therapy is 4000 IU/mL, and it is maintained until no viral load is detected [30].

The study protocol conformed to the ethical guidelines of the 1975 Declaration of Helsinki as reflected in a priori approval by the Ethics Committee for Clinical Research of Cantabria (Internal code: 2020.225). A waiver of informed consent was provided since the study was considered a retrospective review.

### 2.4. Prophylaxis Strategies against Pneumocystis jirovecii in Spanish Liver Transplant Units

All the 25 adult LT units in Spain were asked via email to describe their current prophylaxis strategy against PJP. Specifically, they were asked the following: do you apply a prophylaxis strategy against *Pneumocystis jirovecii* in liver transplant recipients? If so, please detail whether it is universal or in specific cases (must be defined), and specify the drug of choice, dosage, and duration. Otherwise, argue the reasons for not implementing a prophylaxis strategy.

### 2.5. Statistical Analysis

Quantitative variables were expressed as median and interquartile range and qualitative variables as proportions. Cumulative incidence was determined by the number of new PJP cases during the study period divided by the size of the population at risk (i.e., patients transplanted) per 100 (%). Incidence rate of PJP was determined in units of the reciprocal of person transplant years (PTY) calculated up to April 2019, death, or loss to follow-up. Statistical analysis was performed with IBM SPSS Statistics v22.0 for Mac (IBM Corp., Armonk, NY, USA).

## 3. Results

### 3.1. Incidence of Pneumocystis jirovecii in Liver Transplant Recipients

From November 1990 to October 2019, 683 LT procedures were carried out on 631 patients. The most frequent liver disease and indication of LT was alcoholic liver disease and decompensated cirrhosis, respectively. Fifty-two patients were retransplanted and 29 received other transplants, the most frequent of which was combined kidney-liver transplantation (Table 2). Prophylaxis against PJP was established in 21 of these 29 recipients of other additional transplants (20 KT and one bone marrow transplantation) following the corresponding protocols of each program. All of them were given TMP-SMX and none developed PJP. These patients were excluded from the analysis. The other patient who received a bone marrow transplantation died early after the third day and no prophylaxis was undertaken. The reason for not initiating prophylaxis in the remaining KT patients could not be clarified after reviewing the medical records. In the whole LT cohort five cases of PJP were identified, giving an overall cumulative incidence of 0.82% and an incidence rate of 0.99 cases per 1000 PTY.

### 3.2. Clinical Presentation and Outcome of Pneumocystis jirovecii Infection in Liver Transplant Recipients

The risk factors for PJP, clinical features, treatment, and outcome of the five LT patients that developed PJP are shown in Table 3. Of the five patients, only one was diagnosed within the first 6 months post-LT and in two the infection occurred several years after LT. Three cases were diagnosed in the 1990s and had more intense immunosuppressive regimens following the common practice at that time. Pulse steroid therapy for moderate/severe TCMR preceded PJP in two cases and co-existing infections were present in all but one patient. The most frequent symptom and radiological finding were fever with productive cough and ground glass opacities, respectively. In two cases no microbiological confirmation could be achieved, and diagnosis was based on clinical and radiological findings after discarding other aetiologies. In another patient a lung biopsy was needed in order to rule out everolimus-induced interstitial lung pneumonitis. PJP was severe in two patients, causing death in one of them. Except for one, all patients with severe pancytopenia were treated with TMP-SMX.

### 3.3. Pneumocystis jirovecii in Other Solid Organ Transplant Recipients

Table 4 shows the number of transplants, cumulative incidence, and outcome of PJP infection in each type of SOT. KT had the highest cumulative incidence (0.9%). Eight of the 14 KT recipients had been transplanted before the implementation of universal prophylaxis with TMP-SMX for the first 6 months in 1996. In these patients, PJP infection was diagnosed within 6 months in five of them (62.5%). From this period onwards, only one of the 6 cases of PJP (16.7%) was diagnosed within this time frame. Mortality was high regardless of the duration of time since KT. Only one PJP case was identified in LuT and HT, with a cumulative incidence of 0.16% and 0.14%, respectively. The LuT patient received prophylaxis with pentamidine due to sulphonamide allergy. Both cases occurred within the first 6 months and could be successfully treated.

### 3.4. Prophylaxis Strategies against Pneumocystis jirovecii in Spanish Liver Transplant Units

All 25 adult LT units in Spain responded to our query, the majority of which provide PJP prophylaxis (80%). All of these centres reported TMP-SMX as their drug of choice and all use the same dosage—160 mg of TMP and 800 mg of SMX (i.e., double strength) orally three-times weekly. Duration of PJP prophylaxis, however, varied: 12 months (*n* = 4, 16%), 6 months (*n* = 12, 48%), 3 months (*n* = 2, 8%), and between 6 and 12 months (*n* = 2, 8%). These latter two centres maintain prophylaxis for 12 months if steroids are not stopped at 3 months in one centre and at 6 months in the other. Otherwise, prophylaxis is stopped at 6 months. In contrast, five centres (20%) do not indicate prophylaxis. In one of these five centres, TMP-SMX prophylaxis is only applied in patients with human immunodeficiency virus infection and in another centre only if antithymocyte polyclonal antibodies are used (1 case out of the last 200 LT at this centre). All LT units not performing prophylaxis argued a perceived low incidence of PJP at their institution as the primary reason for not employing prophylaxis.

## 4. Discussion

In liver transplant recipients, PJP is most frequently reported before 1992 when immunosuppressive regimens were more intense [8,9]. As these regimens have evolved over time, it is uncertain whether universal PJP prophylaxis is still applicable in the contemporary LT setting. The results of the present study show, in the second largest unprophylaxed LT cohort published to date, a very low incidence of PJP over a 30-year period, with most cases occurring beyond 6 months and during the first decade of the program when higher immunosuppression was prescribed. The survey to LT units in Spain indicates that, while anti-PJP prophylaxis with TMP/SMX is generally implemented in most centres, there is a wide degree of variability within that practice, and there is also an increasing number of centres that do not apply prophylaxis.

The low incidence of PJP in our cohort is in line with recent series in which this infection occurred in less than 3% of LT recipients in the absence of prophylaxis [10,11,12,13,14]. These figures are below the recommended threshold for establishing anti-PJP prophylaxis in SOT patients [1,2], suggesting that previously reported incidence rates, on which the current practice of PJP prophylaxis is based, may have lost validity due to less aggressive immunosuppression regimens and to improvements in the quality of the pre- and post-transplant patient care. It must be highlighted, however, that immunosuppressive regimens vary greatly among centres, with some of them using more intense immunosuppression. Indeed, the use of induction regimens with interleukin-2 receptor blockers and antithymocyte globulins occurs in as many as ~20% and 5% of US liver transplant centres, with ~60% of them applying triple immunosuppression [31]. In contrast, our centre uses less aggressive immunosuppression regimens and, therefore, our findings cannot be extrapolated to centres with higher immunological risk. Two of our cases occurred far beyond the first year which is in agreement with increasing reports of late-onset PJP [24,25]. Both of them had risk factors for its development, which include low total and CD4+ lymphocyte counts, cytomegalovirus infection, hypogammaglobulinemia, graft rejection, and patient age [1,9,24]. In these high-risk patients, many centres tailor PJP prophylaxis by continuing or reinstituting prophylaxis during the period of increased susceptibility [1]. These risk factors, however, do not provide an accurate individual risk assessment and explains why some centres such as our own do not apply prophylaxis even in high-risk patients. In order to decrease the morbidity of this infection but also to avoid unnecessary chemoprophylaxis because of its associated toxicity, well-standardised criteria to establish PJP prophylaxis are most needed. Local PJP prevalence should also be taken into account when assessing this risk, as outbreaks of PJP may occur in nosocomial settings, possibly due to person-to-person spread [1,14]. Our data support a negligible nosocomial transmission at our institution given the absence of outbreaks and the low PJP incidence in the other SOT.

This change in the epidemiology of PJP in LT recipients may lead to different prophylactic strategies among transplant centres. Based on the responses of our survey, there is a lack of consistent or unified approach across LT units in Spain. In line with current guidelines, most of the centres (80%) employ universal anti-PJP prophylaxis, but there is large variability regarding its duration, with a trend towards a shorter period of treatment. This is not surprising, as duration of prophylaxis has relied on expert consensus and not on high-quality evidence [1]. All these centres used the same drug and dosage, TMP-SMX (160 mg/800 mg) three-times weekly. The most striking finding was that 20% of the units do not prescribe prophylaxis due to a perceived low incidence of PJP infection at their institutions. 

The main limitations of our study are related to its retrospective design and to the fact that we do not provide risk factors to better identify patients at high risk for PJP. Our low incidence, however, makes this latter analysis unreliable. Given the thorough examination and the nonrestrictive case definition for PJP (we included patients without microbiological confirmation) we believe in the accuracy of the reported incidence among LT recipients. Nevertheless, we acknowledge that liver recipients who died at other centres would not have been captured in our analysis and that PJP incidence might be underestimated in the other SOT patients, as the identification of PJP cases was based solely on administrative and microbiological records. It must be highlighted, however, that these sources have proved to be acceptably reliable since they identified 80% of PJP cases in LT recipients. Finally, we did not investigate the impact of our strategy on the occurrence of infections caused by other agents sensitive to TMP-SMX. Indeed, TMP-SMX has the potential advantage of being effective at preventing not only other opportunistic infections (e.g., *Toxoplasma gondii* or *Nocardia* spp), but also some respiratory, gastrointestinal, and urinary tract bacterial infections. However, the effectiveness of this additional preventive effect has not been adequately addressed and the routine use of prophylaxis favours the appearance of adverse effects of TMP-SMX. These include increase in serum creatinine, severe hyperkalemia, gastrointestinal complaints, Stevens–Johnson’s syndrome, drug-induced liver injury, interstitial nephritis, and concern for the development of TMP-SMX-resistant *Pneumocystis jirovecii* strains [1].

In conclusion, our findings demonstrate both a low incidence of PJP in our unprophylaxed transplant cohort, with infection occurring in most cases beyond the usual recommended period of prophylaxis, and a significant heterogeneity among prophylaxis strategies across Spanish LT centres. These data do not support a one-size-fits-all approach to PJP prophylaxis and call for new studies that allow for a better characterization of high risk PJP groups in which prophylaxis should be implemented.

## Figures and Tables

**Figure 1 jcm-09-03573-f001:**
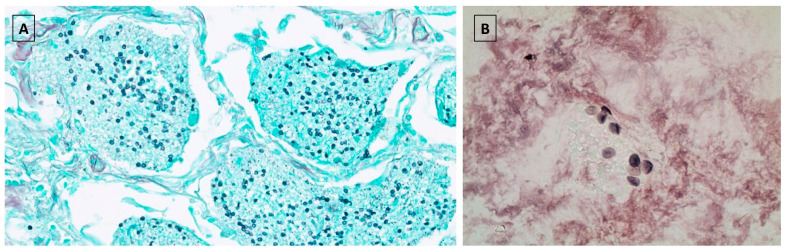
Diagnostic specimens for microbiological demonstration of *Pneumocystis jirovecii* infection. (**A**) Lung biopsy showing intra-alveolar proteinaceous exudates with the presence of numerous *Pneumocystis jirovecii* cysts. Grocott methenamine silver stain (at ×400 magnification). (**B**). Induced sputum showing the presence of numerous *Pneumocystis jirovecii* cysts. Grocott methenamine silver stain (at ×100 magnification).

**Table 1 jcm-09-03573-t001:** Large studies evaluating the incidence of *Pneumocystis jirovecii* pneumonia in liver transplant recipients in the presence or absence of prophylaxis *.

Author and year	*n*	Study Period and Type	Prophylaxis	CI (%)	Mortality (%)	Comments
Kusne et al, 1988 [8]	101	1984–1985 Prospective	No	10.9	27.3	All Cases Occurred Within the First 6 Months and The Three Deaths Had Simultaneous CMV Infection. IR 10 Per 1000 PTY
Hayes et al, 1994 [9]	154	1986–1992 Retrospective	No	5.2	12.5	All Cases Occurred Within the First 6 Months. High-Risk Patients: ≥1 Episode of Rejection, OKT3 Treatment, Or Allograft Dysfunction.
Wade et al, 1995 [10]	284	1990–1993 Prospective	No	0.7	0	Both Cases Occurred Within the First 3 Months.
Hadley et al, 1995 [15]	124	1990–1992 Retrospective	Since July 1991, TMP-SMX q.d.	0	NA	No Prophylaxis Before July 1991
Singh et al, 1997 [16]	130	1989–1995 Prospective	TMP-SMX q.d. indefinitely	0	NA	All Patients Received Tacrolimus as The Primary Immunosuppressive Agent.
Gordon et al, 1999 [17]	265	1987–1996 Retrospective	1987–1991: No	3.8	NS	Cohort Of 1299 SOT Patients. Except For One, All Cases Occurred In the First Year and Without TMP-SMX. IR 3.7 Per 1000 PTY.
			1992–1996: TMP-SMX 1 year			
Torre-Cisneros et al, 1999 [18]	120	NS	TMP-SMX q.d. (*n* = 60)	1.6	0	The Two Cases Occurred in the TMP-SMX Group. No Significant Differences Between Groups. Side Effects In 17–18% In Each Group Without Treatment Discontinuation.
		RCT	SLF-PYT q.w. (*n* = 60)			
Neuman et al, 2002 [19]	646	1988–1995 Retrospective	TMP-SMX t.i.w. until 4 weeks after discharge	1.2	87.5	Splenectomy as A Risk Factor. High Mortality Due to Co-Existing Allograft Dysfunction and CMV Infection. No Case Was on Prophylaxis.
Akamatsu et al, 2007 [20]	180	2000–2003 Prospective	TMP-SMX in 22% guided by BDG levels (>40 pg/mL)	1.1	0	All Living Donor Liver Transplants. Low Positive Predictive Value Of BDG. All Cases Within the First 6 Months. Side Effects Of TMP-SMX In 28%.
Trotter et al, 2008 [21]	853	1997–2007 Retrospective	TMP-SMX t.i.w. (first 3 months)	0	NA	Side Effects Of TMP-SMX Were Not Reported.
Pappas et al, 2010 [22]	378	2001–2006 Prospective	NS	0	NA	Transnet. Data Shown Are from The Surveillance Cohort. Pjp 12-Month Ci of 3% In the Incidence Cohort With 16,808 Sot (4468 Lt).
Orlando et al, 2010 [11]	203	2001–2008 Retrospective	No	0	NA	The Authors Suggested That IS Monotherapy May Nullify the Risk For PCP.
Ohkubo et al, 2012 [23]	156	NS Retrospective	TMP-SMX guided by BDG levels (>40 pg/mL)	2.6	50	All Living Donor Liver Transplants During A 6-Year Period.
Wang et al, 2012 [12]	436	2001–2011 Retrospective	No	1.2	20	All Five Cases Occurred Within the First 7 Months.
Sarwar et al, 2013 [13]	611	2000–2012 Retrospective	No	1.1	71.4	Four of the 7 Cases (57%) Occurred Within the First 7 Months.
Iriart et al, 2015 [24]	345	2004–2010 Retrospective	TMP-SMX t.i.w. the first 6 months	1.4	NS	Case-Control Study. No Case While on Prophylaxis. IR 2.6 Per 1000 PTY. Risk Factors: Age, Lymphocyte Count, And CMV Infection.
Desoubeaux et al, 2016 [14]	285	2011–2014 Retrospective	No	2.1	50	Four Of The Six Cases Occurred During an Outbreak Of PJP. Survival Is Only Reported in These 4 Patients (50%).
Neofytos et al, 2018 [25]	567	2008–2016 Retrospective	354 (62.4%) received prophylaxis	0.7	NS	Swiss Transplant Cohort (2842 SOT). Three Of The 4 Cases in LT Had Received Prophylaxis. Mean Time Post-LT 440 Days (Range 71–1163).

* The minimum number of patients to consider a study as large is 100.Abbreviations: CI, cumulative incidence; CMV, cytomegalovirus; IR, incidence rate; PTY, person transplant year; OKT3, monoclonal antibody targeted at the CD3 receptor; TMP-SMX, trimethoprim-sulfamethoxazole; q.d., daily; NA, not applicable; t.i.w., three times a week; SOT, solid organ transplantation; LT, liver transplantation; SLF-PYT, sulfadoxine/pyrimethamine; q.w., once a week; IS, immunosuppression; BDG, β-D-Glucan; TRANSNET, Transplant-Associated Infection Surveillance Network; PJP, Pneumocystis jirovecii.

**Table 2 jcm-09-03573-t002:** Baseline characteristics of liver transplant recipients.

Variable *	Population (*n* = 610)
Age (Years)	55.3 (48.0–61.1)
Gender (Male)	451 (73.9)
Race (Caucasian)	604 (99.0)
Primary Liver Disease	
Alcohol	280 (45.9)
Hepatitis C	128 (21.0)
Alcohol + Hepatitis C	48 (7.9)
Hepatitis B	36 (5.9)
Primary Biliary Cholangitis	21 (3.4)
Autoimmune Hepatitis	13 (2.1)
Toxic	10 (1.6)
Other	74 (12.1)
Indication of Liver Transplantation	
Decompensated Cirrhosis	332 (54.4)
Hepatocarcinoma	200 (32.8)
Acute Liver Failure	35 (5.7)
Acute-On-Chronic Liver Failure	3 (0.5)
Other	40 (6.6)
Retrasplant	52 (8.5)
Hepatic Artery Thrombosis	14 (26.9)
Recurrence of Primary Liver Disease	10 (19.2)
Biliary Complications	9 (17.3)
Hepatocarcinoma	1 (1.9)
Other	18 (34.6)
Other Transplants	8 (1.3)
Renal (Simultaneous/Consecutive)	5 (0.8)/1 (0.2)
Bone Marrow	1 (0.2)
Heart	1 (0.2)
Death	297 (48.7)
Lost Follow-up **	35 (5.7)
Median Time of Follow-up (years)	6.3 (1.6–12.8)

* Quantitative variables were expressed as median and interquartile range and qualitative variables as absolute value (proportion). ** All these lost were due to change of residence to another region—follow-up was undertaken by the corresponding liver transplant unit.

**Table 3 jcm-09-03573-t003:** Characteristics of patients with *Pneumocystis jirovecii* pneumonia.

Variable	Case 1	Case 2	Case 3	Case 4	Case 5
Age at Diagnosis (Years)/Sex	65.7/Male	51.5/Male	47.4/Male	68.6/Male	69.3/Male
Etiology of Liver Disease	Hepatitis C	Alcohol	Alcohol	Alcohol	Alcohol
Indication Of LT	Hepatocarcinoma	Decomp. Cirrhosis	Decomp. Cirrhosis	Decomp. Cirrhosis	Decomp. Cirrhosis
MELD/Child-Pugh (Points)	11/5	23/9	15/7	14/10	19/10
Year Of LT	1995	1997	1998	2005	2015
Time from LT (Months)	7.6	11.1	3.0	169.4	50.4
Significant Comorbidities	No	No	Psoriasis	Graves´ disease + COPD	Liver Allograft Cirrhosis
D/R CMV Serological Status	D+/R+	D+/R-	D+/R+	D+/R+	D+/R+
Immunosuppression	CsA + Steroids + Azathioprine	CsA + Steroids	CsA + Steroids + Azathioprine	CsA + Everolimus	Tacrolimus + MMF + Everolimus
Acute Rejection Pre-Pneumocystis	No	No	Yes	No	Yes
Treatment of Acute Rejection			Pulses of steroids		Pulses of Steroids
Chronic Rejection	No	No	No	Yes	Yes
Co-Existing Infections	Ophthalmic zoster	CMV	Clostridium difficile	No	SBP
Symptoms					
Fever	Yes	Yes	Yes	Yes	Yes
Cough	Dry	Productive	Productive	No	Productive
Dyspnea	Yes	Yes	No	No	Yes
Thoracic Pain	No	No	No	No	No
Leucocytes (X 10^3/Μ)	5.5	6.2	3.8	6.2	3.0
Lymphocytes (X 10^3/Μ)	0.5	1.5	0.9	2	0.1
Polymorphonuclear (X 10^3/Μ)	4.7	4.1	2.4	3.5	2.5
Chest CT	No	No	Yes	Yes	Yes
Radiological Findings	Ground Glass Opacities	Ground Glass Opacities	Consolidations + Ground Glass Opacities	Consolidations + Ground Glass Opacities	Consolidations + Ground Glass Opacities
Bronchoscopy	No	No	Yes	Yes	Yes
Stain	Positive	Negative	Negative	Negative	Negative
PCR	No	No	No	Positive	Positive
Lung Biopsy	No	No	No	Yes	No
Treatment of Pneumocystis	TMP-SMX + Corticoids	TMP-SMX + Corticoids	TMP-SMX	TMP-SMX + Corticoids	Pentamidine
ICU Admission	Yes	No	No	No	No
Death from Pneumocystis	No	No	No	No	Yes

Abbreviations: COPD, chronic obstructive pulmonary disease; CT, computed tomography; CMV, cytomegalovirus; CsA, cyclosporine; decomp, decompensated; D/R, donor/recipient; ICU, intensive care unit; LT, liver transplantation; MMF, mycophenolate mofetil; PCR, polymerase chain reaction; SBP, spontaneous bacterial peritonitis; TMP-SMX, trimethoprim-sulfamethoxazole;.

**Table 4 jcm-09-03573-t004:** Cumulative incidence of *Pneumocystis jirovecii* pneumonia in other types of solid organ transplantation.

Variables *	Kidney Transplant	Lung Transplant	Heart Transplant
Number of Patients	1600 **	642	705
Number of Transplants	2085	653	720
PJP Cases	14	1	1
Cumulative Incidence (%)	0.88	0.16	0.14
Time from Transplant to PJP Diagnosis (Months)	17.8 (2.0–103.6)	1.5	6.0
PJP Diagnosis Within 6 Months	6 (42.9)	1 (100)	1 (100)
Death Due To PJP	3 (21.4)	0 (0)	0 (0)

* Quantitative variables were expressed as median and interquartile range and qualitative variables as absolute value (proportion). ** Among these, 60 consisted of combined kidney-pancreas transplantation and 26 combined kidney-liver transplantation. Abbreviations: PJP, *Pneumocystis jirovecii* pneumonia.

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
