# Peer review of "Is Routine Prophylaxis Against Pneumocystis jirovecii Needed in Liver Transplantation? A Retrospective Single-Centre Experience and Current Prophylaxis Strategies in Spain"

_jcm, 2020, doi:10.3390/jcm9113573_

Round 1
Reviewer 1 Report
It is a well written manuscript about the need of routine prophylaxis against pneumocystis jirovecii in patients undergoing a liver transplantation
Though this study has a lot of limitations, as it is a retrospective one and it includes results from different liver transplant centers (with different approaches and procedures among them), it is quite well organized and the results are clear.
I have nothing to add or negatively mention.
Author Response
Reviewer 1:
It is a well written manuscript about the need of routine prophylaxis against pneumocystis jirovecii in patients undergoing a liver transplantation. Though this study has a lot of limitations, as it is a retrospective one and it includes results from different liver transplant centers (with different approaches and procedures among them), it is quite well organized, and the results are clear. I have nothing to add or negatively mention.
We would like to sincerely thank the reviewer for his kind comments and acknowledge the limitations he/she mentions.
Reviewer 2 Report
This manuscript is of interest as it addresses the incidence of PJP in a large unprophylaxed cohort of liver transplant recipients. The investigators have retrospectively reviewed the data of 631 patients who underwent LT and no PJP prophylaxis and they reported just 5 positive cases. In addition, they conducted a survey among all 25 Spanish liver transplant centers in order to assess PJP policies.
Part of the difficulty in evaluating this manuscript is that, with due respect for the seniority and international visibility of the authors, the methodology is lacking of some features. Indeed, I think there are some issues here.
1. The authors included in the analysis and in the table 2 also the 21 kidney-liver and BM transplant recipients who actually underwent prophylaxis. Although when calculating cumulative incidence these patients were excluded, I believe they have to be excluded entirely from the analysis itself (including the baseline as shown in table 2) as the aim was to evaluate PJP incidence in LT only with no prophylaxis.
2. In the experimental section of the paper, all the history of the transplant center is out of interest. In addition, the section regarding the incidence of PJP in other organs shouldn't be addressed in this paper. Either the analysis is conducted methodologically in all solid organ transplants (having detailed characteristics and graphs) or LT is done separately.
3. Concerning the survey conducted among Spanish centers: there is no clear explanation regarding the questions and the way data of responses were collected/analyzed.
4. There is no mention in the discussion about the policy of the authors regarding recipients with risk factors. I believe it could be of interested whether the authors would consider PJP prophylaxis in case, for instance, of steroid treatment for rejection or CMV viremia.
Author Response
Reviewer 2:
We would like to sincerely thank all reviewers for their comments and constructive criticisms. We hope to have adequately addressed them
This manuscript is of interest as it addresses the incidence of PJP in a large unprophylaxed cohort of liver transplant recipients. The investigators have retrospectively reviewed the data of 631 patients who underwent LT and no PJP prophylaxis and they reported just 5 positive cases. In addition, they conducted a survey among all 25 Spanish liver transplant centers in order to assess PJP policies. Part of the difficulty in evaluating this manuscript is that, with due respect for the seniority and international visibility of the authors, the methodology is lacking some features. Indeed, I think there are some issues here.
- The authors included in the analysis and in the table 2 also the 21 kidney-liver and BM transplant recipients who actually underwent prophylaxis. Although when calculating cumulative incidence these patients were excluded, I believe they have to be excluded entirely from the analysis itself (including the baseline as shown in table 2) as the aim was to evaluate PJP incidence in LT only with no prophylaxis.
We thank the reviewer for the comment. In the new version of the manuscript we have excluded these patients entirely from the analysis. We have modified the abstract, Table 2 and the remaining results section on this issue accordingly. The latter now reads as follows:
“[…] Prophylaxis against PJP was established in 21 of these 29 recipients of other additional transplants (20 KT and one bone marrow transplantation) following the corresponding protocols of each program. All of them were given TMP-SMX and none developed PJP. These patients were excluded from the analysis. […] In the whole LT cohort five cases of PJP were identified, giving an overall cumulative incidence of 0.82% and an incidence rate of 0.99 cases per 1000 PTY”.
We have also changed the statement that our study represented the largest unprophylaxed LT cohort published to date. After excluding the aforementioned patients, it is the second largest. Indeed, Sarwar et al included 687 LT procedures on 611 patients compared to 652 LT procedures on 610 patients in our cohort.
- In the experimental section of the paper, all the history of the transplant center is out of interest. In addition, the section regarding the incidence of PJP in other organs shouldn't be addressed in this paper. Either the analysis is conducted methodologically in all solid organ transplants (having detailed characteristics and graphs) or LT is done separately.
We have removed the history of the other solid organ transplant programs at our institution. It now reads as follows: “[…] In order to evaluate the local prevalence of PJP infection in other solid organ transplant (SOT) recipients at our institution, a retrospective review regarding this infection was also conducted in recipients of kidney (KT), heart (HT), and lung transplantation (LuT). Universal anti-PJP prophylaxis with TMP-SMX for 6 months is indicated in all KT recipients since 1996, while this prophylaxis is prolonged for life in all LuT recipients since the beginning of the program. In contrast, the HT program has never applied prophylaxis against PJP”.
Regarding the incidence of PJP in the other SOT recipients, we still believe this data is of interest and that it should remain in the manuscript as it evaluates the local prevalence of PJP infection at our institution, including possible outbreaks of PJP that we might have missed. Moreover, our results could have been questioned in case of having found a high incidence of PJP in the other SOT programs. As mentioned in the Discussion, we do acknowledge that PJP incidence might have been underestimated in these SOT patients since the identification of PJP cases was based solely on administrative and microbiological records.
- Concerning the survey conducted among Spanish centers: there is no clear explanation regarding the questions and the way data of responses were collected/analyzed.
We did not send a traditional survey with different questions, each one with closed answers. It was rather an open question asking them to specify whether they applied prophylaxis or not (universal or in specific cases), and in case they did, they had to specify the drug of choice, dosage, and duration.
This section now reads as follows: “All the 25 adult LT units in Spain were queried via email to describe their current prophylaxis strategy against PJP. Specifically, they were asked the following: Do you apply a prophylaxis strategy against Pneumocystis jirovecii in liver transplant recipients? If so, please detail whether it is universal or in specific cases (must be defined), and specify the drug of choice, dosage, and duration. Otherwise, argue the reasons for not implementing a prophylaxis strategy”.
- There is no mention in the discussion about the policy of the authors regarding recipients with risk factors. I believe it could be of interested whether the authors would consider PJP prophylaxis in case, for instance, of steroid treatment for rejection or CMV viremia.
We thank the reviewer for his comment. Our transplant program has never applied PJP prophylaxis in patients being treated for rejection, CMV viremia or in other recipients with risk factors for this type of infection. We have added a comment on this issue in the Discussion. It now reads as follows:
“Both of them had risk factors for its development, which include low total and CD4+ lymphocyte counts, cytomegalovirus infection, hypogammaglobulinemia, graft rejection, and patient age [1,9,24]. In these high-risk patients, many centres tailor PJP prophylaxis by continuing or reinstituting prophylaxis during the period of increased susceptibility [1]. These risk factors, however, do not provide an accurate individual risk assessment and explains why some centres such as our own do not apply prophylaxis even in high-risk patients”.
- The manuscript is well written with the data well presented. In conclusion, the authors called “for new studies that allow for a better characterization of high risk PJP groups in whom prophylaxis should be implemented.” What is the proposal that the authors had in mind: randomized controlled trial? Please elaborate.
We do not suggest conducting clinical trials since the large number of patients that should be included will make this trial unlikely. As mentioned in the Discussion, the current identifiable risk factors for PJP do not provide an accurate individual risk assessment. Therefore, future prospective observational studies should provide new tools (e.g. risk calculators) that will allow a better identification of high-risk LT recipients in whom prophylaxis should be implemented.
- In addition, there is a disconnection in the sentence of the middle of the paragraph below. “The Marques de Valdecilla University Hospital (Santander, Cantabria, Spain) is an urban, academic tertiary care centre with great expertise in organ We conducted a retrospective review regarding PJP infection of all LT performed at our institution since the beginning of our adult LT program in November 1990 to October In the initial years our centre performed all LT, not only from Cantabria, but also from several others Spanish autonomous communities such as Galicia, Basque Country, Canary Islands, Asturias, La Rioja 7 and Castile and Leon. These regions progressively developed their own LT programs over the following 12 years, and since 2009 our program is responsible of all LT performed in Cantabria and La The organ donation activity in these two Spanish autonomous communities is the highest of our country (above 80 donors per million of population), and as of January 1, 2019, their combined population was 895,212. All patients received an ABO-compatible primary orthotopic LT from deceased donors using the piggyback operation [28] and no prophylaxis against PJP was undertaken, except for some patients with combined liver-kidney transplant.” There is a clear disconnection at the line 87. This error needs to be corrected.
We apologize for this error. It now reads as follows: “[…] These regions progressively developed their own LT programs over the following 12 years, and since 2009 our program is responsible of all LT performed in Cantabria and La Rioja. The organ donation activity in these two Spanish autonomous communities is the highest of our country (above 80 donors per million of population) […]”.
Reviewer 3 Report
Drs. Ignacio Fortea et al. present a comprehensive single-center retrospective series from 1990-2019 outlining low PJP incidence (5/631, 0.8% incidence, IR 0.95/1000 pt years, one death) among liver transplant recipients despite no routine utilization of PJP prophylaxis. The report is well-written and is bolstered by several other features: a review of prior large liver series showing a range of PJP incidences, incidence of PJP among other SOT recipients at their center, and an important survey of PJP prophylaxis practices across Spain with a good response rate showing wide variation in use.
To improve the impact and validity of the manuscript, I would recommend the following revisions.
Major comments
Generalizability of findings
Would recommend that the authors speak more to the limitations of generalizability of the findings of low PJP incidence at their center. In particular, it appears that they have an immunologically lower risk center, based upon what seems like uncommon use of antibody induction and frequent use of steroid withdrawal. Contrast this to general US liver transplant practices which appear to utilize anti IL2R induction in ~20% and ATG in 5%, with 60% on triple immunosuppression (doi:10.1111/ajt.15674).
As an addition to Table 2, it would be helpful to see a additional characteristics: proportion of HIV+ recipients, frequency of those given anti IL2 (or ATG) induction, proportion of those on mTOR inhibitors, proportion undergoing steroid withdrawal, frequency of those suffering acute rejection by 1 year post transplant. This was referred to in more narrative form in paragraph 2.3. They did, however, comment that they do not perform ABO incompatible transplants, so I would presume that desensitization is rarely utilized.
Could also comment on CMV prophylaxis protocol (and include the donor recipient status for cases in Table 3), given the well-recognized relationship between CMV and PJP post transplant.
Given the low number of cases, however, formal comparative statistics of recipient factors versus the 5 cases is not needed and wouldn't be particularly meaningful.
The authors do otherwise provide a useful table of prior liver transplant studies showing mostly lower PJP incidences, but as noted with quite varying center characteristics.
Case definitions and understanding robustness of ascertainment:
The authors appeared to use a fairly robust and conservative case definition for PJP. This primarily utilized microbiologic confirmation of diagnosis, though details of standardized testing protocols and which modalities were utilized over time were not explicitly outlined. Otherwise, cases without microbiological confirmation were included if clinical and radiological support were otherwise present, though details of these criteria were not outlined.
To improve the reader's understanding of capture of PJP cases, would recommend briefly, but ideally specifically, describing their criteria for diagnosis of cases in absence of microbiologic diagnosis. Particularly as this was the case for two of their 5 "confirmed" cases!
Additionally, understanding the landscape of their testing practices and capacity during the study period would be helpful. Is PJP testing a standard part of liver transplant recipient bronchoalveolar lavage panels? When did PJP PCR enter clinical practice, versus use of DFA and/or silver staining? Is 1,3-beta d glucan routinely ordered as part of PJP evalution?
Additionally, the authors parsed discharge records for PJP diagnoses using ICD-9 codes. The reported study period is through April 2019, however, which probably overlapped transition to ICD-10 code schemas in the 2015-2016 range. Did the authors search for ICD-10 codes (e.g. B59) or is there crosswalk from ICD-9 built into the medical record?
It would be worth acknowledging that liver recipients who died at other centers would not have been captured in this analysis.
Minor comments
Would expand the risk/benefit discussion touched upon in paragraph 4 of the discussion, namely highlighting that PJP prophylaxis afford many potential "off target" benefits to liver recipients to reduce incidence of listeria, nocardia, toxoplasma, and other bacterial infections. This must be incorporated in the risk/calculus benefit of whether to administer prophylaxis. Additionally, a brief comment on the frequency of adverse drug effects related to TMPSMX would be helpful to again flesh out the risks/benefit discussion of prophylaxis. Similarly commentary has been published by de Boer et al. for kidney recipients (DOI: 10.1111/j.1399-3062.2011.00645.x). I am not immediately familiar with a similar study in liver recipients.
Would also make sure to include commentary that PJP prophylaxis should be tailored post liver transplant as well as following rejection episodes to those with higher risk profiles (antibody induction, lymphodepletion, profund lymphopenia).
Would ideally have supplied the phrasing of the survey tool used to query other transplant centers regarding their PJP prophylaxis approaches and what instrument (if any) was utilized.
Infrequent spelling, capitalization, and grammar issues: eg "linfocytes" in table 3, capitalized P in "progressive" page 5 line 7, page 11 line 143 "TMP-SMF" etc.
Author Response
Reviewer 3:
We would like to sincerely thank all reviewers for their comments and constructive criticisms. We hope to have adequately addressed them
Major comments
Generalizability of findings
- Would recommend that the authors speak more to the limitations of generalizability of the findings of low PJP incidence at their center. In particular, it appears that they have an immunologically lower risk center, based upon what seems like uncommon use of antibody induction and frequent use of steroid withdrawal. Contrast this to general US liver transplant practices which appear to utilize anti IL2R induction in ~20% and ATG in 5%, with 60% on triple immunosuppression (doi:10.1111/ajt.15674).
We thank the reviewer for his comment. We have added a comment on this issue in the Discussion and included the reference provided by the reviewer. It now reads as follows:
“The low incidence of PJP in our cohort is in line with recent series in which this infection occurred in less than 3% of LT recipients in the absence of prophylaxis [10-14]. These figures are below the recommended threshold for establishing anti-PJP prophylaxis in SOT patients [1,2], suggesting that previously reported incidence rates, on which the current practice of PJP prophylaxis is based, may have lost validity due to less aggressive immunosuppression regimens and to improvements in the quality of the pre- and post-transplant patient care. It must be highlighted, however, that immunosuppressive regimens vary greatly among centers, with some of them using more intense immunosuppression. Indeed, the use of induction regimens with interleukin-2 receptor blockers and antithymocyte globulins occurs in as many as ~20% and 5% of US liver transplants centers, with ~60% of them applying triple immunosuppression [31]. In contrast, our centre uses less aggressive immunosuppression regimens and, therefore, our findings cannot be extrapolated to centers with higher immunological risk”.
- As an addition to Table 2, it would be helpful to see additional characteristics: proportion of HIV+ recipients, frequency of those given anti IL2 (or ATG) induction, proportion of those on mTOR inhibitors, proportion undergoing steroid withdrawal, frequency of those suffering acute rejection by 1-year post transplant. This was referred to in more narrative form in paragraph 2.3. They did, however, comment that they do not perform ABO incompatible transplants, so I would presume that desensitization is rarely utilized.
We agree with the reviewer that knowing the additional characteristics he/she mentions would be helpful not only to better describe the profile of the LT recipients from our cohort, but also to identify risk factors for the development of PJP. Unfortunately, we are currently unable to provide these data. Regarding the identification of risk factors, given our low incidence of PJP any attempt to identify risk factors for PJP would not be particularly meaningful or reliable. This is argued both in the Discussion and by the reviewer in question 3.
Although we are unable to provide exact figures of the characteristics asked by the reviewer, we can at least state the following:
- HIV+ recipients represent a very small part of our LT cohort (less than 15 cases).
- We have never used ATG and induction therapy with the interleukin-2 receptor blockers (basiliximab) was used in the last decade as a calcineurin-sparing agent to patients with prior or postoperative significant renal impairment (i.e. creatinine clearance <60 mL/min). Recipients receiving such induction therapy represent less than 15% of our LT cohort.
- Inhibitors of the mammalian target of rapamycin (mTORi) were generally used in case of intolerance to MMF and/or development of de novo malignancy after LT. Their use began in 2008, and once again, represent a very small part of our LT cohort.
- Regarding steroid withdrawal, these drugs were tapered slowly during the first year at the beginning of the program and hereinafter were discontinued 3–6 months post-LT, except for those patients at higher immunological risk (g. immune-mediated diseases such autoimmune hepatitis).
- Our perception is that the proportion of patients suffering acute rejection by 1-year post transplant is similar to previous reported rates (~20-30%) (DOI: 10.1111/j.1600-6143.2012.04140.x)
- Desensitization has never been utilized in our center.
We have added additional information in this regard in the Methods section (2.3. Immunosuppressive drug regimens and cytomegalovirus prophylaxis protocol in liver transplantation).
- Could also comment on CMV prophylaxis protocol (and include the donor recipient status for cases in Table 3), given the well-recognized relationship between CMV and PJP posttransplant. Given the low number of cases, however, formal comparative statistics of recipient factors versus the 5 cases is not needed and wouldn't be particularly meaningful. The authors do otherwise provide a useful table of prior liver transplant studies showing mostly lower PJP incidences, but as noted with quite varying center characteristics.
We have added the CMV prophylaxis protocol in the Methods section (2.3. Immunosuppressive drug regimens and cytomegalovirus prophylaxis protocol in liver transplantation). It now reads as follows:
“As far as the cytomegalovirus (CMV) prophylaxis protocol is concerned, our CMV‐seronegative recipients who receive an organ from CMV‐seropositive donor (D+/R−) receive antiviral prophylaxis with valganciclovir 900 mg po once daily for 3-6 months. This drug is started within the seventh and tenth day after LT. Preemptive therapy with valganciclovir (900 mg po b.i.d. in recipients with normal renal function) is used instead in CMV R+. In these LT recipients CMV viral load (quantitative nucleic acid testing) is measured weekly until discharge and then once every two weeks for the first three months. The viral threshold we use to initiate preemptive therapy is 4000 IU/mL, and it is maintained until no viral load is detected [30]”.
We have also added the donor recipient CMV serological status for the cases in Table 3.
Case definitions and understanding robustness of ascertainment:
- The authors appeared to use a fairly robust and conservative case definition for PJP. This primarily utilized microbiologic confirmation of diagnosis, though details of standardized testing protocols and which modalities were utilized over time were not explicitly outlined. Otherwise, cases without microbiological confirmation were included if clinical and radiological support were otherwise present, though details of these criteria were not outlined. To improve the reader's understanding of capture of PJP cases, would recommend briefly, but ideally specifically, describing their criteria for diagnosis of cases in absence of microbiologic diagnosis. Particularly as this was the case for two of their 5 "confirmed" cases! Additionally, understanding the landscape of their testing practices and capacity during the study period would be helpful. Is PJP testing a standard part of liver transplant recipient bronchoalveolar lavage panels? When did PJP PCR enter clinical practice, versus use of DFA and/or silver staining? Is 1,3-beta d glucan routinely orderedas part of PJP evalution?
We thank the reviewer for his/her comment. We have added a comment on this issue in the Methods (2.2. Cases). It now reads as follows:
“PJP cases were defined by the following criteria: (1) new onset of respiratory symptoms; 2) radiological findings consistent with PJP infection; 3) microbiological demonstration of PJP infection (i.e. Real-time quantitative PCR, and/or Grocott methenamine silver stain performed in samples from bronchial alveolar lavage (BAL), sputum (spontaneous or induced), and transbronchial/open lung biopsy) (Figure 1). Of note, PJP testing has always been systematically performed on BAL from LT recipients. Grocott methenamine silver stain was the standard method used at the beginning of the program, but hereinafter, PJP testing also routinely include PCR. In contrast, (1,3)-β-d-glucan detection is rarely used in the LT setting at our center. Cases without microbiological confirmation were also included if the clinical (fever ± respiratory symptoms) and radiological picture (fine, bilateral, perihilar, diffuse infiltrates that progress to an interstitial alveolar butterfly pattern) supported the diagnosis of PJP”.
- Additionally, the authors parsed discharge records for PJP diagnoses using ICD-9 codes. The reported study period is through April 2019, however, which probably overlapped transition to ICD-10 code schemas in the 2015-2016 range. Did the authors search for ICD-10 codes (e.g. B59) or is there crosswalk from ICD-9 built into the medical record?
We thank the reviewer for his/her comment. This section has been modified and now reads as follows:
“3) Hospital discharge records. The latter consisted of a list of all patients admitted to our hospital with diagnosis upon discharge of PJP registered as code 136.3 of the International Classification of Diseases, Ninth Revision, Clinical Modification (ICD-9-CM), and as code B59 or J17.3 of the ICD-10 (it replaced ICD-9-CM since January 2016)”.
- It would be worth acknowledging that liver recipients who died at other centers would not have been captured in this analysis.
We agree with the reviewer and we have included this limitation on the Discussion. It now reads as follows:
“Nevertheless, we acknowledge that liver recipients who died at other centers would not have been captured in our analysis and that PJP incidence might be underestimated in the other SOT patients as the identification of PJP cases was based solely on administrative and microbiological records”.
Minor comments
- Would expand the risk/benefit discussion touched upon in paragraph 4 of the discussion, namely highlighting that PJP prophylaxis afford many potential "off target" benefits to liver recipients to reduce incidence of listeria, nocardia, toxoplasma, and other bacterial infections. This must be incorporated in the risk/calculus benefit of whether to administer prophylaxis. Additionally, a brief comment on the frequency of adverse drug effects related to TMPSMX would be helpful to again flesh out the risks/benefit discussion of prophylaxis. Similarly commentary has been published by de Boer et al. for kidney recipients (DOI: 1111/j.1399-3062.2011.00645.x). I am not immediately familiar with asimilar study in liver recipients.
As the reviewer suggested, we have expanded the risk/benefit discussion on TMP-SMX prophylaxis. It now reads as follows:
“Finally, we did not investigate the impact of our strategy on the occurrence of infections caused by other agents sensitive to TMP-SMX. Indeed, TMP-SMX has the potential advantage of being effective at preventing not only other opportunistic infections (e.g. Toxoplasma gondii or Nocardia spp), but also some respiratory, gastrointestinal and urinary tract bacterial infections. However, the effectiveness of this additional preventive effect has not been adequately addressed and the routine use of prophylaxis favors the appearance of adverse effects of TMP-SMX. These include increase in serum creatinine, severe hyperkalemia, gastrointestinal complaints, Stevens-Johnson’s syndrome, drug-induced liver injury, interstitial nephritis, and concern for the development of TMP-SMX-resistant Pneumocystis jirovecii strains [1].
- Would also make sure to include commentary that PJP prophylaxis should be tailored post liver transplant as well as following rejection episodes to those with higher risk profiles (antibody induction, lymphodepletion, profund lymphopenia).
We thank the reviewer for his comment. Our transplant program has never applied PJP prophylaxis in patients being treated for rejection, CMV viremia or in other recipients with risk factors for this type of infection. We have added a comment on this issue in the Discussion. It now reads as follows:
“Both of them had risk factors for its development, which include low total and CD4+ lymphocyte counts, cytomegalovirus infection, hypogammaglobulinemia, graft rejection, and patient age [1,9,24]. In these high-risk patients, many centres tailor PJP prophylaxis by continuing or reinstituting prophylaxis during the period of increased susceptibility [1]. These risk factors, however, do not provide an accurate individual risk assessment and explains why some centres such as our own do not apply prophylaxis even in high-risk patients”.
- Would ideally have supplied the phrasing of the survey tool used to query other transplant centers regarding their PJP prophylaxis approaches and what instrument (if any) was utilized.
We did not send a traditional survey with different questions, each one with closed answers. It was rather an open question asking them to specify whether they applied prophylaxis or not (universal or in specific cases), and in case they did, they had to specify the drug of choice, dosage, and duration. It was sent via email.
This section now reads as follows: “All the 25 adult LT units in Spain were queried via email to describe their current prophylaxis strategy against PJP. Specifically, they were asked the following: Do you apply a prophylaxis strategy against Pneumocystis jirovecii in liver transplant recipients? If so, please detail whether it is universal or in specific cases (must be defined), and specify the drug of choice, dosage, and duration. Otherwise, argue the reasons for not implementing a prophylaxis strategy”.
- Infrequent spelling, capitalization, and grammar issues: eg "linfocytes" in table 3, capitalized P in "progressive" page 5 line 7, page 11 line 143 "TMP-SMF" etc.
We apologize for these errors that have been corrected.
Reviewer 4 Report
In this manuscript that is titled “Is Routine Prophylaxis Against Pneumocystis Jirovecii Needed in Liver Transplantation? A Retrospective Single-Centre Experience and Current Prophylaxis Strategies In Spain”, José Ignacio Fortea et al report an evidence that routine prophylaxis for Pneumocystis jirovecii pneumonia (PJP) might not be needed.
From 1990 to 2019, their center performed 683 liver transplants in adult patients and found that a cumulative incidence and incidence rate of 0.82% and 0.99% with 106 cases per 1000 PTY, respectively among those who did not receive the prophylaxis against PJP. Among other solid organ transplants at their center, the authors found cumulative incidences of 0.88%, 0.16%, and 0.14% among 2085 KT, 653 lung transplants, and 720 heart transplants, respectively. KT had the highest cumulative incidence (0.9%) and mortality. In addition, the authors performed a survey with 25 adult LT centers in their country. Their practice without routine PJP prophylaxis is supported by 20% of the surveyed programs that do not use prophylaxis.
The manuscript is well written with the data well presented. In conclusion, the authors called “for new studies 197 that allow for a better characterization of high risk PJP groups in whom prophylaxis should be implemented.” What is the proposal that the authors had in mind: randomized controlled trial? Please elaborate. In addition, there is a disconnection in the sentence of the middle of the paragraph below.
“80
- The Marques de Valdecilla University Hospital (Santander, Cantabria, Spain) is an
- urban, academic tertiary care centre with great expertise in organ We conducted
- a retrospective review regarding PJP infection of all LT performed at our institution since the
- beginning of our adult LT program in November 1990 to October In the initial years our
- centre performed all LT, not only from Cantabria, but also from several others Spanish
- autonomous communities such as Galicia, Basque Country, Canary Islands, Asturias, La Rioja
7 and Castile and Leon. These regions
- Progressively developed their own LT programs over the following 12 years, and since 2009 our
- program is responsible of all LT performed in Cantabria and La The organ donation activity in
- these two Spanish autonomous communities is the highest of our country (above 80 donors per
- million of population), and as of January 1, 2019, their combined population was 895,212
- All patients received an ABO-compatible primary orthotopic LT from deceased donors using the
- piggyback operation [28] and no prophylaxis against PJP was undertaken, except for some patients with combined liver-kidney transplant.”
- There is a clear disconnection at the line 87. This error needs to be corrected.
Author Response
Reviewer 4:
In this manuscript that is titled “Is Routine Prophylaxis Against Pneumocystis Jirovecii Needed in Liver Transplantation? A Retrospective Single-Centre Experience and Current Prophylaxis Strategies in Spain”, José Ignacio Fortea et al report an evidence that routine prophylaxis for Pneumocystis jirovecii pneumonia (PJP) might not be needed. From 1990 to 2019, their center performed 683 liver transplants in adult patients and found that a cumulative incidence and incidence rate of 0.82% and 0.99% with 106 cases per 1000 PTY, respectively among those who did not receive the prophylaxis against PJP. Among other solid organ transplants at their center, the authors found cumulative incidences of 0.88%, 0.16%, and 0.14% among 2085 KT, 653 lung transplants, and 720 heart transplants, respectively. KT had the highest cumulative incidence (0.9%) and mortality. In addition, the authors performed a survey with 25 adult LT centers in their country. Their practice without routine PJP prophylaxis is supported by 20% of the surveyed programs that do not use prophylaxis.
- The manuscript is well written with the data well presented. In conclusion, the authors called “for new studies that allow for a better characterization of high risk PJP groups in whom prophylaxis should be implemented.” What is the proposal that the authors had in mind: randomized controlled trial? Please elaborate.
We do not suggest conducting clinical trials since the large number of patients that should be included will make this trial unlikely. As mentioned in the Discussion, the current identifiable risk factors for PJP do not provide an accurate individual risk assessment. Therefore, future prospective observational studies should provide new tools (e.g. risk calculators) that will allow a better identification of high-risk LT recipients in whom prophylaxis should be implemented.
- In addition, there is a disconnection in the sentence of the middle of the paragraph below. “The Marques de Valdecilla University Hospital (Santander, Cantabria, Spain) is an urban, academic tertiary care centre with great expertise in organ We conducted a retrospective review regarding PJP infection of all LT performed at our institution since the beginning of our adult LT program in November 1990 to October In the initial years our centre performed all LT, not only from Cantabria, but also from several others Spanish autonomous communities such as Galicia, Basque Country, Canary Islands, Asturias, La Rioja 7 and Castile and Leon. These regions progressively developed their own LT programs over the following 12 years, and since 2009 our program is responsible of all LT performed in Cantabria and La The organ donation activity in these two Spanish autonomous communities is the highest of our country (above 80 donors per million of population), and as of January 1, 2019, their combined population was 895,212. All patients received an ABO-compatible primary orthotopic LT from deceased donors using the piggyback operation [28] and no prophylaxis against PJP was undertaken, except for some patients with combined liver-kidney transplant.” There is a clear disconnection at the line 87. This error needs to be corrected.
We apologize for this error. It now reads as follows: “[…] These regions progressively developed their own LT programs over the following 12 years, and since 2009 our program is responsible of all LT performed in Cantabria and La Rioja. The organ donation activity in these two Spanish autonomous communities is the highest of our country (above 80 donors per million of population) […]”.
Round 2
Reviewer 2 Report
I have read all the comments with interest and I would like to thank the Authors for their exhaustive responses. All the comments were addressed in a structured way and the manuscript was amended accordingly.
I believe that the overall quality of the manuscript is improved.
Many thanks to all other reviewers for their valuable comments.